# Effect of the Partial Replacement of Cement with Waste Granite Powder on the Properties of Fresh and Hardened Mortars for Masonry Applications

**DOI:** 10.3390/ma15249066

**Published:** 2022-12-19

**Authors:** Zuzanna Zofia Woźniak, Adrian Chajec, Łukasz Sadowski

**Affiliations:** Department of Materials Engineering and Construction Processes, Wrocław University of Science and Technology, Wybrzeże Wyspiańskiego 27, 50-370 Wroclaw, Poland

**Keywords:** granite powder, waste, cement mortar

## Abstract

Granite is a well-known building and decorative material, and, therefore, the amount of produced waste in the form of granite powder is a problem. Granite powder affects the health of people living near landfills. Dust particles floating in the air, which are blown by gusts of wind, can lead to lung silicosis and eye infections, and can also affect the immune system. To find an application for this kind of waste material, it was decided to study the effect of partially replacing cement with waste granite powder on the properties of fresh and hardened mortars intended for masonry applications. The authors planned to replace 5%, 10%, and 15% of cement with waste material. Series of mortar with the addition of granite powder achieved 50% to 70% of the compressive strength of the reference series, and 60% to 76% of the bending strength of the reference series. The partial replacement of cement with the granite powder significantly increased the water sorption coefficient. The consistency of the fresh mortar, and its density and water absorption also increased when compared to the reference series. Therefore, Granite powder can be used as a partial replacement of cement in masonry mortars.

## 1. Introduction

The constantly developing construction sector results in a variety of masonry structures, among others, being made [1]; therefore, there is a need to modify traditional masonry mortars. One of the factors influencing the development of the construction industry is the market demand for finding a use for waste materials. Another factor that affects the development of the construction industry is the desire to carry out work as quickly and correctly as possible [2]. Additionally, there is also a need to develop materials that have a variety of properties. Despite the rapid development of the construction industry and the creation of new building materials, there is also a problem with regards to the correct execution of cement mortars [3].

The main purpose of masonry mortar is to permanently connect masonry elements. The mortar must have the correct consistency, and a good compressive and bending strength. A mortar should be selected according to the masonry element and the weather conditions in which the masonry works are conducted. The masonry mortar affects the capillary rise in the wall in which it is used; therefore, the intended use of the brick object is also important. The choice of components of masonry mortars, which are partially replaced with waste materials, is not accidental. In many studies, waste materials are added to mortars as a substitute for cement or aggregate, which is a solution to the global problem associated with these two materials. Using waste materials as substitute materials for cement reduces the costs related to, for example, the storage of such waste materials, and is an appropriate method of their disposal [4,5]. Many articles over the past few years have been devoted to the study of carbon dioxide emissions. The current challenge, due to the continuous emission of carbon dioxide into the atmosphere, is to take care of the environment. Cement production produces carbon dioxide CO_2_, which is responsible for 5% of the total anthropogenic carbon dioxide emissions [4], with some sources even giving a value of 7% [6] or 8% [7]. Over the past few years, cement plants have been modernized in order to reduce the CO_2_ emissions resulting from cement production. However, the problem has not been completely resolved. Therefore, scientists are constantly conducting more studies that aim to reduce or optimize the amount of used cement. In cement mortars, it has been found to be possible to replace cement with various waste materials. Even a small replacement of cement with waste material can have a positive impact on the environment [8,9].

Many countries have a problem finding a way to use waste materials in the construction industry [10]. There is an interest among scientists in waste materials, such as granite powder, which shows that there is a problem related to the use of this material [11,12,13]. Granite is an igneous rock, the hardness of which is estimated to be 7 on the Mohs scale. This material, as a rock, is a very durable material that does not dissolve in water. Therefore, granite powder is a mineral material of natural origin. This material is found on virtually every continent, even in Antarctica. In Poland, the biggest granite deposits are located in Lower Silesia [14]. The extraction of granite in Europe is counted in millions of tons per year. Granite and marble are the second most frequently mined non-metallic minerals [15], and are the most frequently used decorative materials [16]. They are used for the production of tombstones, road surfaces, sidewalks, and fences. The processing of these materials generates by-products, such as granite powder and coarse aggregate. Granite powder has sharp edges due to cutting processes. The shape of granite powder particles differs from the shape of cement particles, therefore such added waste material fills in the aggregate skeleton. The experimental part of this work is to investigate the effect of granite powder on the selected properties of mortars. Currently, granite powder is not a commonly used material in construction, but it is used, for example, as a fertilizer for plants. It can also be used instead of sand to fill paving stones. However, the amounts of granite powder waste indicate that there is an insufficient use of this waste. There is a problem with the disposal of such granite powder, because it is a non-biodegradable material [17], and often ends up in landfills. Piles of material pose a threat to human life due to gusts of wind that carry away their particles. A person exposed to a long-term contact with crystalline silicon dioxide SiO_2_ is at risk of lung silicosis. People staying or living in the vicinity of dust dumps containing SiO_2_ have problems with their immune systems, and are exposed to all kinds of lung changes [18]. Granite powder also causes eye infections due to air pollution [17].

This article aims to investigate the effect of granite powder on selected properties of cement mortars. The focus was on properties, such as wall capillary suction, water absorption, temperature, and strength. All these properties affect the integrity of the joint between the mortar and the structural element. These properties correlate with the factors to which the mortar is exposed in its normal working conditions. In their research, the authors tried to find a possibility of using waste material, such as granite powder, in mortars, which would result in less cement being produced for this purpose. This will in turn contribute to the reduction of carbon dioxide emissions. The replacement of a few percent of cement with another material may have a small effect in reducing the amount of carbon dioxide that is emitted into the atmosphere. However, when taking into consideration the entire production of cement that is used in mortars, such a replacement can significantly reduce the amount of carbon dioxide emissions. The conducted research is very important due to the possibility of improving the quality of life of people and the environment. The article describes an innovative approach to environmental protection.

## 2. Materials and Methods

This point has been divided into several points. In Section 2.1, the materials from which the mortar series were created are described.

### 2.1. Materials Used in the Research

In the research, Portland Cement type II 32.5R (Odra in Opole, Poland), aggregate (Górażdże, Poland), granite powder (Strzegom, Poland), and tap water were used. The particle size distribution of the used granite powder is shown in Table 1. The chemical compositions are provided in Table 2. The grain size of the aggregate was 0–1.8 mm. The aggregate was dried for one week at a temperature of 25 °C.

The chemical compositions are provided in Table 2. The grain size of the aggregate was 0–1.8 mm. The aggregate was dried for one week at a temperature of 25 °C.

Four cementitious mortar compositions were investigated. The binder of the reference series was made just with Portland Cement type II 32.5R. Three series were made by replacing 5%, 10%, and 15%, respectively, of the mass of the cement with granite powder. The components of the mix used for the tests came from the same batch of products. The materials were stored at a constant temperature. The cement, aggregate, and granite did not have any lumps. The prepared proportions of the components are presented in Table 3.

Mixing took place using a mortar drill, and began with placing the aggregate, cement, and granite powder in measured amounts into a mixer. After about 30 s, the ingredients were mixed manually to eliminate fractionation. Then, three-quarters of the total mass of water was added while mixing continuously with a mortar drill for approximately 30 s. Afterwards, manual mixing was conducted in order to clear the bottom of the container of leftover unmixed ingredients. One-quarter of the total water was added during the mixing. The final mixing was for about one minute.

### 2.2. Properties of the Fresh Mortars

Immediately after the completion of the mixing process, the temperature of the fresh mortar was examined. A pin thermometer was used to test the temperature of the mix. The thermometer measures the temperature within the range from −50 °C to +300 °C, with an accuracy of ±1 °C, and a resolution of 0.1 °C. The temperature was measured every half an hour, starting from immediately after mixing to 90 min after the end of the mixing process. The consistency of the evaluated series was measured using a flow table in accordance with PN-EN 1015-3 [19]. The test equipment includes a flow table, truncated cones, and a compactor. The test involved determining the spreading diameter of the mortar on a flow table. The bulk density was evaluated by measuring the weight of a 1000 cm^3^ container filled with mortar in two layers. Each layer was vibrated in order to remove any air bubbles.

### 2.3. Properties of the Hardened Cement Mortars

The following sub-sections describe how the tests were conducted on hardened samples of the masonry mortars.

#### 2.3.1. Bending Strength and Compressive Strength

After determining the bulk density of the fresh mix, samples with dimensions of 40 × 40 × 160 mm were taken for the bending strength tests. Before the filling moulds, their inner surface was covered with a release agent. The hardened series were demoulded after 3 days, and kept in an air temperature of 20 ∓ 2 °C. The bending strength tests were conducted in accordance with PN-EN 196-1 [20] after 28 days of the samples maturing. The value of the bending strength was obtained based on the arithmetic mean of the three evaluated samples, which was rounded up to 0.01 MPa.

After the bending strength tests, the compressive strength tests were conducted in accordance with PN-EN 1015-11 on six halves of 40 × 40 × 160 mm beams, which were obtained after the bending strength tests [21].

#### 2.3.2. Water Absorption

Water absorption is a measure of the amount of water being absorbed into a material, and has an influence on the capillary suction of a wall [20]. Moisture mainly migrates up a masonry wall through mortar. Mass water absorption is determined from Equation (1):
(1)$$n_{w}=\frac{m_{n}-m_{s}}{m_{s}}\;100,\;\left(\%\right)$$
where: $n_{w}$—mass water absorption (%);$m_{n}$—mass of the material sample when saturated with water (kg);$m_{s}$—mass of a sample of the material in its dry state (kg).

First, the samples were marked, and a scale was drawn on them (a line in the middle of the sample). The samples were then weighed with an accuracy of 1 g and placed on washers in a vessel so that they would not touch the bottom of the vessel. The washer consisted of two layers of 1 mm thick plastic mesh. The vessel was filled with water to half the height of the samples. The water temperature was 18 ± 2 °C. After 24 h, the samples were wiped and then weighed on scales with an accuracy of 1 g. After that, water was added to the level of 10 mm above the surface of the sample. After 24 h, the weight of the samples was evaluated until the weight of the saturated samples was constant.

#### 2.3.3. Capillary Suction

Capillary suction of a wall leads to an increase in the moisture content of the wall, and thus to imperfections in the appearance of the building [22]. It affects the thermal insulation of the wall and the durability of wooden elements. The capillary suction tests involved recording the capillary rate by measuring the mass of the samples. In the first period of time, the amount of absorbed water by the samples was greater, which was also shown in the studies of Halina Garbalińska et al. [23]. The measurements continued until a constant weight of all the series was achieved-with a deviation of 1%. Therefore, it was decided to introduce five phases of measuring the mass of the samples. The time distances between individual measurements were as follows:
Phase I lasted 5 h and measurements were taken every hour;Phase II lasted 8 h and measurements were taken every 2 h;Phase III lasted 24 h and measurements were taken every 4 h;Phase IV lasted 16 h and measurements were taken every 8 h;Phase IV lasted 36 h and measurements were taken every 12 h.

The research was conducted until a constant weight was obtained for each series. The main aim of the test was to calculate the water sorption coefficient using Equation (2):
(2)$$A=\frac{∆m_{t}}{F\;∆\sqrt{t}}$$
where: A—water sorption coefficient [kg/(m^2^h^1/2^)];$∆m_{t}$—increase in the sample’s mass (kg);F—suction surface (m^2^);$∆\sqrt{t}$—increment of the root of time (h^1/2^).

A plastic grate was placed on the bottom of a transparent plastic container. The samples were weighed and numbered, then placed on a 3.5 cm thick plastic grate. Water was gradually added so that the samples were immersed to a height of 5 mm. The water level was kept constant up to the immersion height of the samples of 5 mm. The samples were weighed until moisture equilibrium was reached.

## 3. Results

### 3.1. Temperature

The temperature of the mixes affects the time of their transport, the properties of the obtained hardened mortars [24], and the maintenance of the properties of the fresh mortars, e.g., their workability. A comparison of the temperature test results for the GP series and the REF series is presented in Figure 1.

The temperature for all the cases decreases over time. The highest temperature difference during 90 min was achieved by the reference series, and amounted to 6.4 °C. The GP5 series had a difference of 3.6 °C. The GP10 and GP15 series obtained the smallest temperature differences in the evaluated time, amounting to 3.2 °C and 2.8 °C, respectively. It can be concluded that with an increasing amount of granite powder added to the mix, the temperature difference, measured at 90 min intervals, decreases. Immediately after the end of the mixing process, the temperature for the series with the addition of granite powder varied by several degrees. The largest temperature difference in the first measurement was over 5 °C between the REF and GP5 series, with the reference series having the highest temperature of 22 °C in the first measurement. The biggest difference in the last measurement was 2.4 °C between the GP10 and GP5 series. The results of the last measurement were similar for the REF, GP10, and GP15 series. The difference between the GP15 and REF series and the GP15 and REF series was 0.2 °C. The differences in temperature for these series after 90 min from the end of the mixing process did not show big differences. This means that replacing the cement with granite powder in the amount of 10% and 15% of the cement mass will not significantly affect the temperature of the mix within 30 min to 90 min from the end of the mixing process. The exception was the GP5 series, for which the mix temperature was the lowest. Such a temperature may have a positive effect on concreting in summer conditions. The decrease in the temperature is due to the reduction in the amount of pozzolanic material. Pozzolanic material reacts as soon as it comes into contact with water. Replacing cement with a material with different pozzolanic properties causes the temperature to drop. Z. Woźniak et al. [24] noticed a similar trend regarding a drop in temperature for the series with the addition of granite powder when compared to the reference series. The authors measured the temperature immediately after mixing, then again 30 min after the end of the mixing process. They examined mortar series with a 10%, 15%, and 20%, respectively, replacement of cement with granite powder. In the case of the second measurement (after 30 min), there was a clear relationship between the increasing amount of granite powder in the binder and the decreasing temperature. Another mineral material that influences the hydration temperature is limestone powder. Grzeszczyk et al. [25] claimed that the addition of lime powder in mortar reduces the heat of hydration.

### 3.2. Consistency of the Mix and Its Bulk Density

Results of the consistency tests based on two measurements for each series are presented in Figure 2a, and results of bulk density are presented in Figure 2b. The results of the series with a partial replacement of cement with granite powder (green columns) were compared with the results obtained for the reference series (grey column). Error bars indicate the standard deviation from the mean value for the evaluated series.

The addition of the granite powder waste material significantly increased the consistency of the mix when compared to the reference series. As the amount of waste material in a mix increases, the flow increases. The largest flow was achieved for the GP15 series, and it was 30 mm larger than that of the REF series. This difference is about 16% of the flow of the REF series. Prokopski et al. [26] examined the consistency of concrete mixes with the addition of granite powder using the slump method. He achieved a similar effect-the consistency of mixes increased with the addition of granite powder. The authors suggest that this effect is related to the addition of material with a smaller grain size than that of cement and sand. The addition of granite powder increased the packing density and decreased the porosity of the cement. Granite powder particles in the form of dust surround cement particles and cause the filling in of voids and the reduction in the friction between the particles.

The partial replacement of cement with granite powder resulted in an increase in density when compared to the reference series. As the amount of granite powder increases, the bulk density increases. The highest density among all the evaluated series was achieved by the GP10 series with a 10% replacement of cement with granite powder. Granite has a smaller fraction than the used cement. Replacing cement with material with smaller fractions will increase the packing density, which will in turn affect the bulk density. The addition of the granite powder caused the filling of voids between the sand and cement, which increased the bulk density. Similar results were obtained by the authors of article [24], in which granite powder caused a higher volume density of the mix when compared to the series without the additive. Woźniak and Chajec [24] observed that it may be related to the filling in of the aggregate skeleton with granite powder, which has less particles than cement. Article [11] also investigated the effect of adding granite powder to the cement mass on the bulk density, and it was observed that the partial replacement of cement with granite powder increased the bulk density of the mortar. Four series were made, in which 10%, 20%, and 30%, respectively, of the cement was replaced with granite powder. The highest density was achieved by the series with the 10% replacement of cement. It was observed that with the increasing amount of granite powder in the series, the bulk density of the mix increased.

### 3.3. Bending Strength and Compressive Strength

Figure 3a presents the bending strength, and Figure 3b shows the compressive strength. The error bars refer to the standard deviation from the average values for the results obtained by the series.

Based on Figure 3a, it can be concluded that the partial replacement of cement with granite powder reduces the bending strength. The reference series achieved the highest bending strength. The second-highest compressive strength was achieved by the GP10 series. The lowest values were achieved by the GP15 series. For the evaluated series, there is no relationship between the amount of cement replaced with a given waste material and the bending strength. However, it can be concluded that the partial replacement of cement with the granite powder significantly reduced the bending strength. The most optimal amount of replaced cement with the materials used is 10% by weight of cement.

Similar values were achieved by Woźniak et al. [24], where the relationship between a decreasing bending strength and an increasing amount of granite powder in the binder was obtained. The authors evaluated the bending strength after 28 days of the maturation of mortars, which were prepared with a 10%, 20%, and 30%, respectively, replacement of cement with granite powder, and were named GP10, GP20, and GP30, respectively. The difference between the GP15 and the REF series amounted to 44.78% of the compressive strength of the REF series. In the case of the GP10 series, this difference was equal to 30.27% of the REF series. A similar result, but lower than that of the GP10 series, was achieved by the GP5 series, with its compressive strength being equal to 20.54 MPa. Ramadji et al. [13] also investigated the effect of the compressive strength of mortars prepared with the replacement of cement with granite powder in amounts of 10%, 15%, 20%, respectively, of the weight of CEM I cement. It was noticed that with the increase in the amount of granite powder, the compressive strength decreased after 7, 14, and 28 days. After 90 days, the difference between the series with the 10% replacement of cement with granite powder, and the REF series was only 0.1 MPa. This difference was equal to 0.2% of the strength value of the REF series. Nascimento et al. [27] evaluated the compressive and tensile strength of mortars with a partial replacement of cement or aggregate with granite powder. The series with the cement replacement achieved lower strengths than the reference series, which they claim is due to the reduction of the amount of the binder. Higher strengths were obtained for the series with sand replacement, which is due to an increase in the packing density, an increase in the specific weight, and greater resistance.

### 3.4. Water Absorption

Figure 4 presents the results of water absorption tests for the reference series (grey column), and the series with a partial replacement of cement with granite powder (green columns). Error bars indicate the standard deviation from the mean value for the evaluated series.

There is a tendency of increased water absorption with an increased amount of waste material in the mortar. The GP5 series had the lowest water absorption. The series with a 15% replacement of cement with granite powder had the highest water absorption value, which was higher than that of the REF series by 1.51%. Gupta et al. [17] claim that water absorption is inversely proportional to mechanical resistance. Therefore, the highest water absorption was achieved for the series with the lowest compressive strength (GP15).

Similar results were obtained by Chajec et al. [12], who evaluated the absorbability of cement screeds placed on a concrete substrate. This type of flooring is mainly used in housing construction. He evaluated four series of samples, where granite powder was used as a cement substitute in amounts of 10% (GP10), 20% (GP20), and 30% (GP30), respectively, of the cement mass. The values for the series with additives were compared to the REF series. The samples were subjected to two different curing conditions. Some of the samples were stored in air at 20 °C and 50% humidity, and some were subjected to wet curing, where the temperature was 24 °C and the humidity was 85%. The authors noted that the curing conditions had a significant effect on the water absorption, which was evaluated after 90 days of maturation. For the series hardened in the air, there was a tendency of increased water absorption with the addition of granite powder. In turn, there was a reverse tendency for the series cured in wet conditions. It was found that the partial replacement of cement with granite powder did not reduce water absorption if appropriate hardening conditions were maintained.

A similar Influence of granite powder on water absorption in cement mortar was also observed by Alok Damare et al. [28]. They used granite powder as a partial replacement of sand in amounts of 2.5%, 5%, 7%, 10%, 12.5%, 15%, 17.5%, and 20%, respectively, of the sand’s weight. Dry samples were placed in a water container for 48 h, then the amount of absorbed water was measured in percent by weight. The series with the 5% and 7.5% sand replacement with granite powder showed a reduced water absorption. The 2.5% sand replacement series showed the same absorbance as the reference series. The remaining series showed a tendency of increased water absorption with increasing amounts of granite powder in the mortar.

### 3.5. Capillary Suction

Measurements of the mass of the samples, which were planned at equal time intervals divided into five phases, allowed for the creation of graphs showing the process and speed of capillary rising for each of the series. The planned time of the measurements simulates the mortar being exposed for several days to atmospheric factors such as rain, snow, dew, and high air humidity. This situation also reflects the threat to walls caused by flooding or an increase in the level of groundwater. Figure 5 shows the dependence of the change of ∆mt/F over time for the reference series, and for the series with the partial replacement of cement with granite powder.

Based on the mass measurements during the test, it can be concluded that for the period up to 29 h, there is a tendency for higher water absorption with an increasing amount of granite powder in the mortar. The differences between the REF, GP10, and GP10 series are slight, while the GP15 series is significantly different from the other series. The differences between the GP15 series and the other series reached about 1300 kg/m^2^. On the basis of Figure 6, the water sorption coefficient “A” was determined, which is the slope of the straight section of the function $f\left(t^{0.5}\right)=∆m_{t}/F$. These rectilinear sections are presented in the graphs below. The same time intervals were selected (with three measurements), and, based on them, linear trend lines with equation descriptions were created in Figure 6. Error bars indicate the standard deviation from the mean value for the evaluated series.

Based on the directional coefficients of the linear trend lines, Figure 7 was created in order to show the water sorption coefficient.

The capillary suction varies significantly between the evaluated series. The series with a partial replacement of cement with granite powder had higher water sorption coefficients than the reference series. The highest coefficients out of all the GP series were achieved by the GP10 and GP15 series. Their results differed by only 2.09 kg/m^2^h^0.5^. The high sorption coefficients of the GP series may be due to the fact that the surface of the cement particles is very rough, and the particles of granite powder, which are smaller, penetrate into the uneven surface of the cement particles. Prokopski et al. [26] tests were conducted on cubic samples with dimensions of 10 × 10 × 10 cm. They found that the samples made on the basis of CEM I 32.5 R cement had a higher water absorption than the series based on CEM I 42.5 R. They also showed that the addition of granite reduces the wicking capacity. The difference may result from the used test method, because the water was fed at a pressure of 500 kPa, and the penetration was measured in mm. The mass measuring method used in the conducted tests is more accurate due to the fact that it is the amount of water that is absorbed into the sample that is measured. Due to this, the total amount of water, not only the water contained on the surface, can be determined. Additionally, in the case of the samples with dimensions of 4 × 4 × 16 cm, the differences in the absorbed water across their width may be smaller than in the case of the wider samples with dimensions of 10 × 10 × 10 cm. The level of water contained in the centre of a larger sample may be much lower than for smaller samples. The surface of the samples is also important because the capillary suction is different in the case of sample that have surfaces of different roughness.

## 4. The Importance of the Research

The cost of cement mortar with addition of granite powder, could be reduced. Another advantage of using this granite powder could be a reduction in the cost of storing this material. The quality of the storage could also be improved, which would in turn reduce the amount of materials exposed to wind that can transport it to neighbouring settlements. Replacing cement with a different material could reduce the price of mortar. Such a mortar could be used, for example, in dry places due to its increased sorption when compared to traditional mortar. Mortar with the addition of granite powder could be used in walls that are not subjected to significant bending and compressive loads. However, further studies are needed to verify the potential application of such a mortar.

## 5. Conclusions

The conducted studies filled the gaps in the knowledge related to the dependence of ∆ mt/F. The positive effect of partially replacing cement with granite powder on the temperature of fresh cement mortar, and its water absorption, was described. Based on the presented study, the following main conclusions can be drawn:

Reducing the amount of pozzolanic material resulted in a decrease in the temperature of the fresh mortar. This property could enable masonry works to be conducted in summer conditions. The addition of a material with a different shape of grains causes an increase in the flow of a mix and changes in its specific density, which in turn affects the properties of hardened mortars.

The creation of innovative masonry mortars may be associated with the deterioration of their mechanical properties-replacing cement with granite powder for the evaluated series led to a decrease in compressive strength and bending strength. Among the evaluated series, the optimal value of replacing cement with granite powder is 5% to 10% of the weight of cement. Changes in the mechanical properties for these series are acceptable due to the positive aspects associated with the use of granite powder.

To sum up, an appropriate selection of the optimal amount of granite powder in mortar may lead to such a mortar being used. The desired properties of mortars depend on the type of construction material and their exposure to weather conditions during setting and use. Future research should focus on investigating other properties, such as the setting speed with regards to mortar temperature, or the mortar’s adhesion to bricks. In addition, further research is needed in order to estimate the exact amount of cement that can be replaced with granite powder, which in turn would contribute to the development of the practical application of the newly designed mortar. Moreover, subsequent research would bring scientists closer to both the implementation of the proposed solution and the creation of a ready-made mix of mortar ingredients.

## Figures and Tables

**Figure 1 materials-15-09066-f001:**
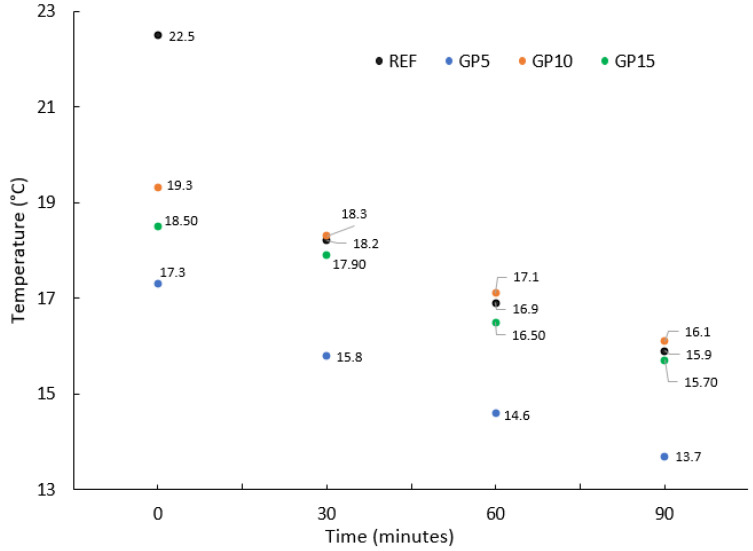
The temperature of the series of the cement mortar evaluated half an hour after the end of the mixing process and up to 2 h after the end of the mixing process.

**Figure 2 materials-15-09066-f002:**
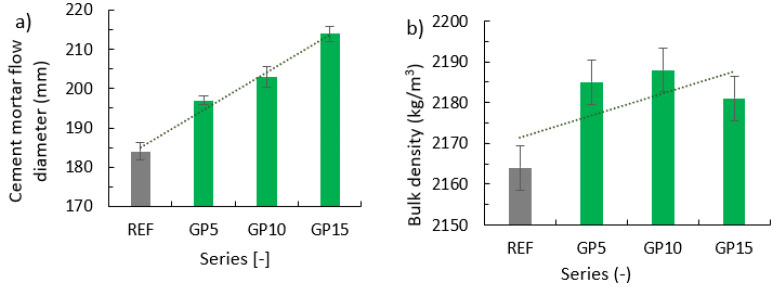
Properties of the fresh cement mortars with a partial replacement of cement with the granite powder: (**a**) cement mortar flow diameter; (**b**) bulk density.

**Figure 3 materials-15-09066-f003:**
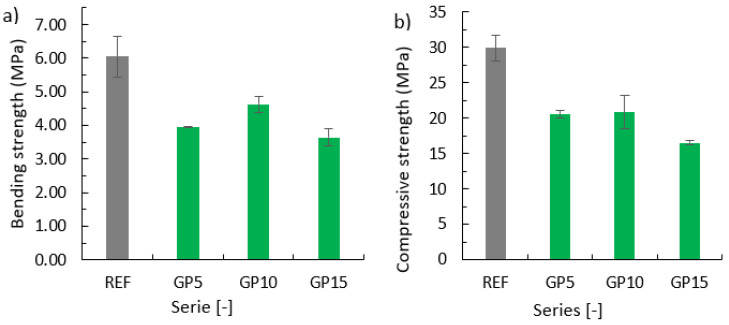
Properties of the hardened cementitious mortars with a partial replacement of cement with the granite powder: (**a**) bending strength; (**b**) compressive strength.

**Figure 4 materials-15-09066-f004:**
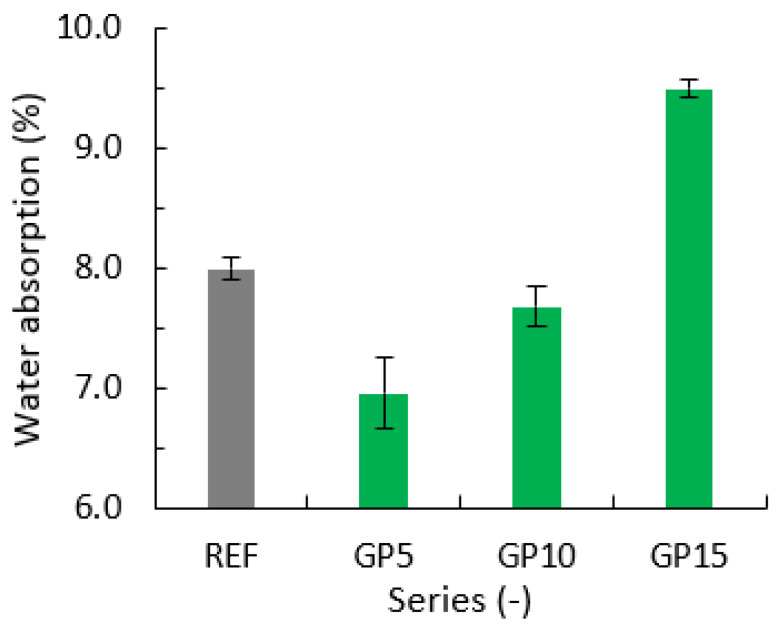
Water absorption for the reference series and series with a partial replacement of cement with granite powder (GP).

**Figure 5 materials-15-09066-f005:**
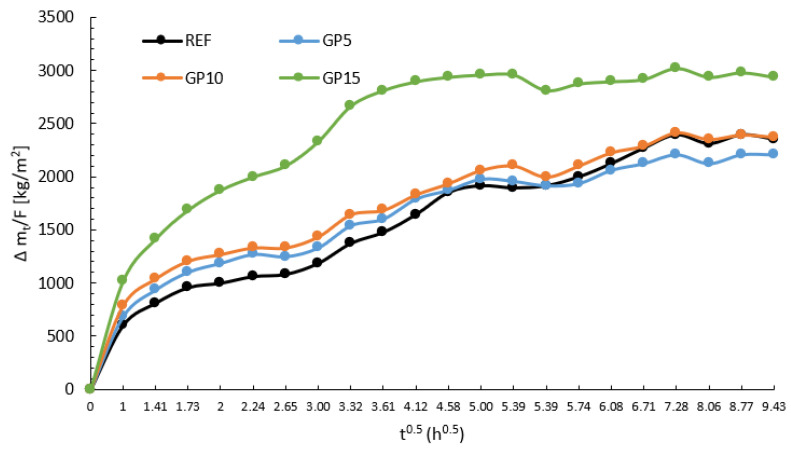
Diagram of ∆mt/F as a function of time root t^0.5^ for the series with the partial replacement of cement with granite powder and the reference series.

**Figure 6 materials-15-09066-f006:**
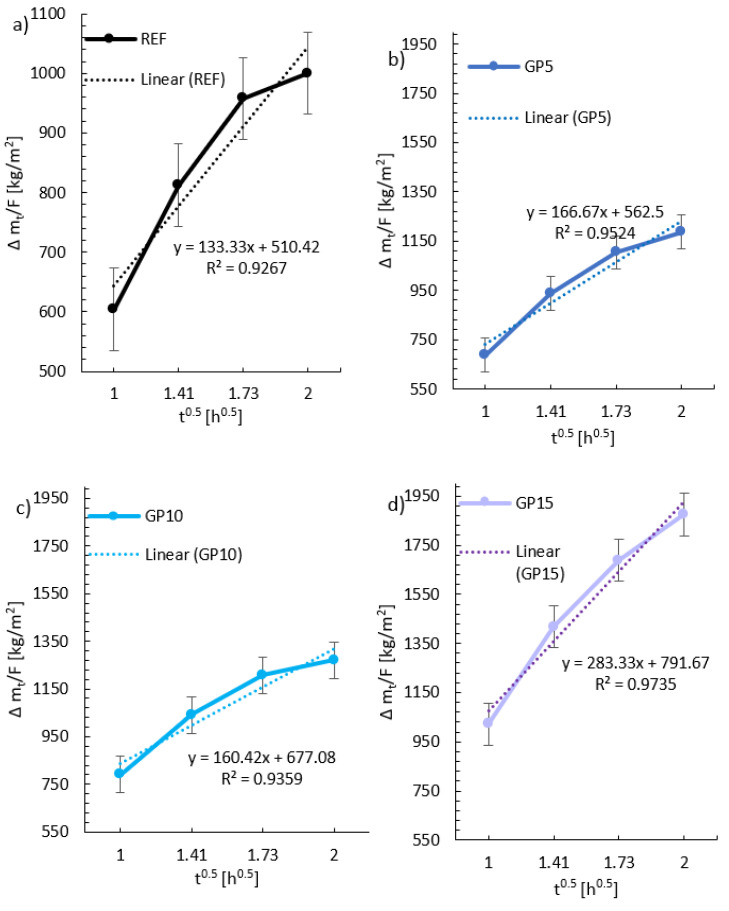
Graphs of capillary suction for the three measuring points with a linear trend line for: (**a**) the reference series; (**b**) the series with the 5% replacement of cement with granite powder; (**c**) the series with the 10% replacement of cement with granite powder; (**d**) the series with the 15% replacement of cement with granite powder.

**Figure 7 materials-15-09066-f007:**
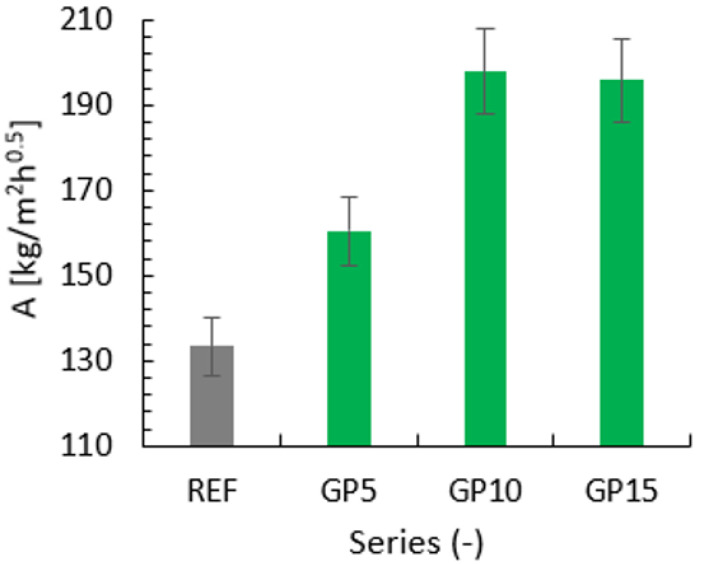
Water sorption coefficients for the reference series (REF) and the series with the partial replacement of cement with granite powder (GP).

**Table 1 materials-15-09066-t001:** Graining curve of the waste material used in the research.

Sieve Size (mm)	<0.02	0.02	0.032	0.063	0.1	0.14
Passes through a sieve (%)	0	5	51.9	82.9	90.2	100

**Table 2 materials-15-09066-t002:** Chemical composition of the used granite powder (%).

Chemical Compound	SiO_2_	Al_2_O_3_	Fe_2_O_3_	CaO	MgO	FeO	NaO	K_2_O
Granite powder (%)	54	20	2	8	6	4	3	3

**Table 3 materials-15-09066-t003:** The proportions of the components of the designed cement mortar mixes.

Series (-)	Portland Cement (%)	Granite Powder (%)	Aggregate (%)	Water (%)	Water/Cement Ratio	Water/Binder Ratio
REF	21.45	-	64.34	14.21	0.66	0.66
GP5	20.38	1.07	64.34	14.21	0.70	0.66
GP10	19.30	2.14	64.34	14.21	0.74	0.66
GP15	18.23	3.22	64.34	14.21	0.78	0.66

## Data Availability

The data presented in this article are available within the article.

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
