# Peer review of "Effect of the Partial Replacement of Cement with Waste Granite Powder on the Properties of Fresh and Hardened Mortars for Masonry Applications"

_materials, 2022, doi:10.3390/ma15249066_

Round 1

Reviewer 1 Report

The authors presented an article titled: “Effect of partial replacement of cement with waste granite powder on the properties of fresh and hardened mortars for masonry applications”. The authors have shown in this paper the potential use of the granite powder as the partial replacement of cement in masonry applications. The article can be interesting from an engineering point of view, however there are many technical mistakes in the article that require further explanation.

Comment 1:

Abstract

Abstract must be improved. Demonstrate in the abstract novelty, practical significance. Add quantitative and qualitative work results to the abstract.

Comment 2:

2. Materials and Methods

Enter the introductory sentence in chapter 2, before subchapter 2.1.

Enter the introductory sentence in subchapter 2.3, before subchapter 2.3.1.

Add the numbering of the formulas in the subchapters: 2.3.3, 2.3.4.

Correct the numbering of subchapter 2.3.3 (it should be subchapter 2.3.2).

Correct the numbering of subchapter 2.3.4 (it should be subchapter 2.3.3).

Comment 3:

3. Results and discussion

Correct the numbering of chapter 2 (it should be chapter 3).

Enter the introductory sentence in chapter 3 (now it is incorrectly marked in text  as 2), before subchapter 3.1.

Improve the quality of Figure 2.

Correct the numbering of subchapter 2.2 (it should be subchapter 3.2).

Correct the caption under the drawing Figure 3.

Correct the numbering of subchapter 2.3 (it should be subchapter 3.3).

Correct the numbering of subchapter 2.4 (it should be subchapter 3.4).

Correct the numbering of subchapter 3.7 (it should be subchapter 3.5).

Comment 4:

6. Conclusions

This chapter must be improved. Add quantitative and qualitative work results. In addition, it is necessary to more clearly show the novelty of the article and the advantages of the proposed method. What is the difference from previous work in this area? Show practical relevance. Presented conclusions are only a description of the test results. Conclusions should reflect the purpose of the article.

Comment 5:

The article must be revised in accordance with the guidelines for authors of Materials journal. There is no numbering of text lines, which makes it difficult to point out errors in the article. The article must be proofread by a native English speaker. The quality of all figures, tables and equations in the text is very poor. The entire article needs to be improved in the numbering of chapters and subchapters. These errors evidence of the carelessness of the authors in writing a scientific article. Literature must be revised and written in accordance with the guidelines for authors. The article can be interesting and helpful but authors must carefully study the comments and make improvements to the article step by step. Mark all changes in color. After major changes the article can be considered for publication in the “Materials” journal.

Author Response

Thank you for all the good remarks in this review. Also, we appreciate critical opinions. We hope that all your contributions improved the quality of this paper.

Reviewer 2 Report

"Effect of partial replacement of cement with waste granite powder on the properties of fresh and hardened mortars for masonry applications" The article is interesting. A few observations are given below, 

1) The abstract is not clear. An abstract is a short summary of your completed research. It is intended to describe your work without going into great detail. Abstracts should be self-contained and concise, explaining your work as briefly and clearly as possible. Please take some time to revise and rearrange the abstract to highlight the overall of your research. 

2) For readers to quickly catch the contribution of this work, it would be better to highlight major difficulties and challenges, and the authors' original achievements to overcome them, in a clearer way in the Introduction section.

Also, the recent literature available in this field should be included in the introduction section. A few examples are given below for your reference,

"Amin, M. N., Khan, K., Saleem, M. U., Khurram, N., & Niazi, M. U. K. (2017). Aging and curing temperature effects on compressive strength of mortar containing limestone quarry dust and industrial granite sludge. Materials, 10(6), 642." 

"Shah, M. I., Amin, M. N., Khan, K., Niazi, M. S. K., Aslam, F., Alyousef, R., ... & Mosavi, A. (2021). Performance evaluation of soft computing for modeling the strength properties of waste substitute green concrete. Sustainability, 13(5), 2867."

3) Please edit your table No1, the image quality is not acceptable. 

4) Please number the equations, and equations must be quoted with proper references. Further, list all of the abbreviations and symbols clearly. 

5) Authors must summarize results in a more systematic way with reference to the previous studies.

6) Conclusions are too limited to prove the significant outcome of this study. Please highlight your results and present them in a systematic way.

Author Response

(The authors gave the same response as above.)

Reviewer 3 Report

In this paper, the author investigates the effect of granite powder on selected properties of cement mortars. The focus was on properties such as wall capillary suction, water absorption, temperature and strength. To sum up, by appropriate selection of the optimal value of granite powder in the mortar, this material can find its application. The whole article has only basic discussion and methods, but it is more like a report when we see innovation. It is suggested to condense the innovation points and highlight the key points.

The abstract does not discuss the results, only the purpose and method

1.Page 2, method of disposal [4], . This place is punctuated incorrectly

2.1/4, the writing is correct?

3. In Figure 3, the gray line indicates that the data looks a little confused

4. In Figure 6, the color is similar, please replace.

Author Response

Thank you for all the good remarks in this review. Also, we appreciate critical opinions. We hope that all your contributions improved the quality of this paper. The whole article has been included in the MDPI template, including the literature, so we hope that after the changes the article is more readable.

Reviewer 4 Report

The conducted work “Effect of partial replacement of cement with waste granite powder on the properties of fresh and hardened mortars for masonry applications” is good. However, following comments should be addressed to further improve paper:

A. GENERAL COMMENTS FOR PAPER ON OVERALL BASIS

1.      The numbering of headings and sub headings should be corrected.

2.      Plain version of paper should have been submitted. Paper with comments is not good to submit in a journal.

3.      Explicitly mention the novelty and research significance of current work in last paragraph of introduction section with emphasis on scientific soundness. Also, add recent relevant literature review more from 2022 papers in introduction section as there are only three papers cited from 2022.

4.      Avoid paragraph of few (2-4) sentences throughout the manuscript.

5.      Results are explained in a descriptive way, thus results in current form look like a project report. Results should be further elaborated with scientific reasoning.

6.      A separate brief section (explaining the relevance of this research for practical implementation) may be added before conclusion section.

7.      Conclusions are little long; these should be to the point as obtained from results. 

8.      English Language should be improved throughout the manuscript.

B. SPECIFIC COMMENTS FOR IMPROVING FOCUSSED RESEARCH

1.      There should be an in-depth discussion about optimized mix for masonry application as claimed in title.

2.      The impact of fresh properties on hard properties should also be discussed.

Author Response

(The authors gave the same response as above.)

Round 2

Reviewer 1 Report

The reviewer would like to thank authors for all the corrections made to the manuscript. Authors should once again check and correct the list of References (for example Reference 25 - line 504). The text of the article still  needs native speaker (English) corrections. After these corrections the reviewed article may be published in Materials journal.

Author Response

Thank you very much for your attention regarding the references. The reference has been corrected. We attach Certificate of proofreading.

Reviewer 3 Report

It looks good after editing.

Author Response

Thank you for accepting the article and valuable comments.